# Is the ENaC Dysregulation in CF an Effect of Protein-Lipid Interaction in the Membranes?

**DOI:** 10.3390/ijms22052739

**Published:** 2021-03-08

**Authors:** Birgitta Strandvik

**Affiliations:** Department of Biosciences and Nutrition, Karolinska Institutet NEO, 14183 Stockholm, Sweden; birgitta.strandvik@ki.se; Tel.: +46-70-5486348

**Keywords:** CFTR, linoleic acid, renal excretion, sodium, sweat, supplementation

## Abstract

While approximately 2000 mutations have been discovered in the gene coding for the cystic fibrosis transmembrane conductance regulator (CFTR), only a small amount (about 10%) is associated with clinical cystic fibrosis (CF) disease. The discovery of the association between CFTR and the hyperactive epithelial sodium channel (ENaC) has raised the question of the influence of ENaC on the clinical CF phenotype. ENaC disturbance contributes to the pathological secretion, and overexpression of one ENaC subunit, the β-unit, can give a CF-like phenotype in mice with normal acting CFTR. The development of ENaC channel modulators is now in progress. Both CFTR and ENaC are located in the cell membrane and are influenced by its lipid configuration. Recent studies have emphasized the importance of the interaction of lipids and these proteins in the membranes. Linoleic acid deficiency is the most prevailing lipid abnormality in CF, and linoleic acid is an important constituent of membranes. The influence on sodium excretion by linoleic acid supplementation indicates that lipid-protein interaction is of importance for the clinical pathophysiology in CF. Further studies of this association can imply a simple clinical adjuvant in CF therapy.

## 1. Introduction

In cystic fibrosis (CF), the latest development of modulators, correctors and potentiators, of the defective gene product—the cystic fibrosis transmembrane conductance regulator (CFTR)—has created important opportunities to influence the defective protein and its functions [1,2]. However, there are still barriers to influencing many different mutations, and to explaining many of the symptoms, inspiring other therapeutic alternatives, such as gene therapy, RNA repair or replacement, and other chemical interferences.

The high level of inflammation has been found to precede the infections [3,4,5,6,7], which further will increase inflammation creating a vicious circle. Inflammation has been associated with lipid abnormality [8,9,10,11], including high prostanoid production [12,13,14,15]. Impact on the symptomatology has also been linked to the overactivity of the epithelial sodium channel (ENaC), highly expressed in lungs, colon, female reproductive tract and vas deferens, kidneys, and some exocrine glands, like the sweat gland [16,17,18]. ENaC facilitates sodium absorption across apical epithelial cell membranes, and its hyperactivity in CF has been suggested to strongly contribute to the abnormal sticky secretion [19,20]. This impact of ENaC on CF pathophysiology is stimulating the development of modulators to inhibit the increased activity of this channel [21,22]. CFTR has its function in lipid structures, and lipid abnormalities have been associated with the disease for more than 50 years [23], but the interest to find treatment modalities in this field has not been encouraged. However, in recent years an increasing interest has focused on the bidirectional relation between proteins and lipids [24,25,26], also in relation to CFTR [27,28,29,30], and lately, the interaction between the modern CFTR modulators and membrane lipids have been shown [31].

## 2. ENaC in CF

### 2.1. ENaC

ENaC is composed of three subunits, the ENaC-α, ENaC-β, and ENaC-γ, all necessary for a functional channel. Each subunit consists of two transmembrane helices and an extracellular loop. In the pancreas, testes, and ovaries, a fourth unit, ENaC-δ, can replace the α-unit, and form a functional channel. The three units build a central ion pore [16]. An animal model with overexpression of the ENaC-β subunit developed CF-like lung disease, indicating that hyperabsorption alone creating airway dehydration can induce the CF phenotype [32]. Overexpression of the β-unit, in the absence of CFTR dysfunction, has also been shown to increase NLRP3-mediated inflammation, indicating that the inflammatory responses in CF might be exaggerated by dysregulated ENaC-dependent signaling [33]. This increasing knowledge of the importance of ENaC hyperactivity has encouraged the search for modulators to inhibit ENaC activity, especially focusing on the pulmonary tract [34]. However, the membrane localization of the protein might open for other possibilities to influence its function.

### 2.2. CFTR and ENaC

The association between CFTR and ENaC has developed the hypothesis that CFTR is directly influencing the ENaC activity, but the limitations in technology have also generated questions about how close the locations of the proteins might be [35,36,37]. In rats, both alveolar cells I and II expressed CFTR and ENaC, but with the dominance of β- and α-subunits of ENaC in type I cells and of CFTR in type II cells [38]. In one study using high-resolution immunofluorescence on mice airways and female reproductive tract, CFTR and ENaC were located on different parts of the cell. The location of ENaC, but not of CFTR, was along the length of the cilia, and CFTR was located on the apical membrane outside of the cilial borders [17]. The results suggest that the influence of a defective CFTR cannot be directly linked to ENaC, but might be transformed via some factor, either created by the abnormal CFTR, like the modulated chloride excretion, or by a factor influencing both channels without direct contact between them. Interaction of other proteins, like aquaporins or Na/K-ATPase, have been discussed to contribute to the disturbed ion balance, but cannot explain the interaction between ENaC and CFTR. Of interest is the study showing that correction of the CFTR activity does not correct the hyperactive ENaC in CF cells [39].

Both CFTR and ENaC are intra-membranous channels, and as such, are influenced by the surrounding membrane constitution, its phospholipid signatures [40,41], its sphingolipids/ceramides [42], as well as its cholesterol content [43]. Membrane lipids are important for raft formation, especially the interplay between cholesterol and docosahexaenoic acid, and between phosphatidylserine and ceramides determining transport in/of proteins in the membrane [44]. Phosphatidylcholine (PC) is an important constituent in cell membranes [45], mostly expressed in the outer membrane layer, and has an increased turnover in CF [46,47]. Phosphatidylserine (PS) is an important phospholipid in the inner membrane layer and interacts with sphingolipids, like ceramides, for cell signaling and transmembrane transports, and has been shown to stabilize the function of CFTR [48]. All these types of lipids have been found disturbed in different cell systems in CF, and would, thus, be potential confounders for the channel activities.

ENaC can be regulated by cytoplasmic Ca^2+^, extracellular ions and pH, and phosphoinositids (PI). PI are important lipids in cell signaling, in shaping membranes, controlling trafficking, and in organelle physiology [49]. The most common fatty acids in PI are stearic acid (18:0) and arachidonic acid (AA, 20:4n-6). It is also suggested that AA-rich PI are the source of phospholipase A_2_ mediated AA release, which is the rate-limiting step in the prostanoid synthesis. Phosphatidylinositol 4,5-bisphosphate (PI(4,5)P_2_) is possibly interacting directly with the channel influencing its gating [50,51,52]. Olivenca et al. [53] have suggested that the regulation of PI(4,5)P_2_ is dependent on balance in the production of phosphatidic acid from PC, which is interesting, since that would relate to the increased PC turnover in CF [46,47].

Ivacaftor (VX-770), and lumacaftor/ivacaftor (VX-809/VX-770), and the triple combination (VX-661/VX-445/VX-770) have all been shown to influence lipid signatures in blood or bronchial epithelial cells, respectively [31,54,55]. All tested combinations of modulators in immortalized bronchial cells markedly influenced lipids by both up- and downregulations. Especially many species of ceramides and PC were downregulated, but PIs were not investigated in their system [55]. In the context of the extent influences on lipids by the modulators, they summarize: “Quite surprisingly, membrane lipid therapy has never been proposed for CF” [55], which is not quite true, when studying the extent of the literature of linoleic acid supplementation. However, randomized, double-blind studies are hitherto lacking.

## 3. Lipid Abnormality in CF Related to ENaC

### 3.1. Membrane Lipid Abnormality

The most consistent lipid abnormality in CF is the decreased concentration of linoleic acid (LA, 18:2n6), known for more than 50 years [23], and shown in all different blood components, plasma and serum, red blood cells, platelets, lipoproteins, triacylglycerides, cholesteryl esters, phospholipids and in different tissues, such as nasal tissue, liver, lungs, heart, muscle, and adipose tissue (for review see Reference [56]). This deficiency is associated with type of mutation, and seems to be more expressed in patients with more severe phenotype [57], but it is also present to some degree in patients without pancreatic insufficiency [58,59], and in heterozygotes, as parents to patients [60,61], suggesting that only severe disturbance is associated to clinical symptoms. LA is an essential fatty acid and an important constituent in cell membranes and is transformed to AA, which is the substrate for the pro-inflammatory eicosanoid system. The AA release by phospholipase A_2_ is increased in CF [62,63,64,65], and since this release is the rate-limiting step in the eicosanoid synthesis, prostanoids are increased [12,13,14,15], a factor which would explain the basic inflammation of the disease. Infection further triggers the AA cascade and the prostanoid system, enhancing the inflammation. This series of events has raised a theoretical fear to increase the inflammation by providing LA [66]. However, many studies show that supplementation with LA does not increase AA [67,68,69,70,71], neither an increase of inflammatory mediators [72], indicating some yet undetermined regulatory mechanism, which also acts in CF [11].

Many studies with supplementation of LA have been reported after 1975, when a case was reported about a newborn, who under treatment with Intralipid™ (rich in LA), regained the pancreatic function [73]. This good effect on the pancreas was not repeated in other cases, but many studies with LA supplementation showed improvement in clinical status, growth, and pulmonary function [74,75,76,77,78]. The moderate effect in many of these studies might relate to too short intervention time and/or too low a dose of supplementation, as illustrated in one long-term study for three years [79], where an interruption immediately showed a decrease in blood lipids with an increase when the supplementation was restarted [80]. It is also important to consider that the plasma concentration does not always reflect the tissue levels [81]. In an animal model of LA deficiency, some tissues seemed more sensitive to deficiency than others [82], and also, the tissue accumulation at supplementation was more directed to some organs [83], indicating different lipid metabolism in different tissues, which is not easily followed in clinical studies. Of special interest are some results of the clinical LA supplementation on sodium homeostasis, which will therefore be reported in more detail.

### 3.2. Linoleic Acid Supplementation and ENaC

Three long-term supplementation studies of LA (12 months) have shown a decrease in sweat sodium without influence on the chloride concentrations [13,74,84]. The decreases were continuous during the study time and resulted in up to a 30% decrease in sweat sodium during one year. Unfortunately, most supplementation studies have only focused on the chloride concentrations, which have only been influenced by some of the modulator therapies, acting directly on the CFTR channel.

Other interesting results about sodium homeostasis were seen in the studies of kidney function in CF patients, showing a very low renal sodium excretion with two different methods [85,86]. These results are not exceptional because many others have reported similar results, but with totally different interpretations [87]. The common view has been that low renal sodium is a result of the increased sweat chloride concentration, i.e., a way of the body to save sodium, and therefore, resulted in the general recommendation of sodium supplementation. In Sweden, the interpretation is the opposite, that the increased renal reabsorption makes extra sodium unnecessary, except at heat exhaustion, similar to for healthy people. The reason for this interpretation is the results of studies, which tried to overcome a potential deficiency by both giving very high loading of sodium and also by preparing the patients by very high oral loading for five days before sodium loading [85]. Both regimens showed similar low renal excretion compared to healthy individuals, suggesting increased unaffected high sodium reabsorption. This would also fit very well with an increased ENaC activity in the kidneys, but the ENaC was not discussed in these studies, because ENaC was not defined at that time. Interestingly, the sodium excretion improved and even normalized in some cases after three years of LA supplementation, and also the glomerulus filtration normalized [80]. An investigation of total body sodium homeostasis with ^24^Na did not indicate a high sodium loss in the patients [88], but might be related to the fact that these patients were already on LA supplementation, being a regular treatment of those patients for at least a decade [89,90]. The results might, thus, raise the question if the patients really have a salt deficiency, when the ENaC in both airways and kidneys are hyperactive. The normal blood concentration of sodium might be an effect of (potentially compensatory) high sweat excretion. There would have to be extremely high sweat production (which is not obvious when performing sweat tests) to believe that the increase of ENaC in airways and kidneys are compensation for sweat loss of sodium. The Swedish patients in the two biggest CF centers have not been recommended sodium supplementation, not even when staying in the Mediterranean, without any reported problems.

The Swedish policy at the centers in Stockholm and Gothenburg, supervising about 65% of the Swedish CF patients have been given regular doses of LA orally and parenterally with Intralipid™, in connection with intravenous antibiotics, since the 1970s and 1990s, respectively. The lung function has been very good, also when compared with other Nordic countries with similar socioeconomic standards, but with much higher antibiotic treatment [78,90,91,92]. In the context of the renal influence by the LA supplementation, it cannot be excluded that the ENaC in the airways have been modified during this regimen, explaining why better lung function was associated with less need for antibiotics, although this is difficult to prove [78,90,92]. Nevertheless, the LA supplementation as a routine has not increased inflammation and elastase, and TGF-α have not been found to increase in the patients (Wretlind and Strandvik, unpublished observation). This routine supplementation used for decades must be considered safe [89,90].

The results on sodium excretion by LA supplementation might reflect an improvement in the membrane structure and thereby influencing ENaC, and open for a membrane-lipid therapy as a cheap adjuvant treatment in CF. The development of new drugs by high-throughput techniques is extremely expensive, restricting the availability in low and middle-income countries to gain from these signs of progress [93]. The modulators are also not without adverse effects [94], and long-term side effects are unknown. The influence of correctors and potentiators on basic mechanisms cannot exclude unwanted effects, including epigenetic mechanisms, if given in early life. LA supplementation might balance the possibility that the ENaC abnormality might be more important for developing severe phenotypes than the CFTR dysfunction. Still, the mechanisms for the lipid abnormality have to be further studied, because what is the primary link to CFTR in the disturbed lipid pattern of phospholipids, ceramides, and cholesterol is not known.

## 4. Conclusions

Recent studies have implied that ENaC hyperactivity has a great impact on CF pathophysiology. Some studies show an influence on sodium excretion both in sweat and kidneys by LA supplementation. This suggests that the lipid-protein interaction in membranes might be of importance for the abnormal function of the channel. It will be interesting to follow the sodium excretion in the double-blind, randomized supplementation of LA for one year, provided as a multicenter European study (ClinTrial.gov NCT 04531410). Such a study is necessary for general recommendation of LA supplementation as an adjuvant therapy to the modulators, or in some cases, as a membrane-lipid therapy alternatively to traditional therapy only focusing on protein interference.

## Data Availability

The original data about cytokines (Wretlind and Strandvik) can be obtained from the author. All other data presented are published and referred to in the references.

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
