# Peer review of "Is the ENaC Dysregulation in CF an Effect of Protein-Lipid Interaction in the Membranes?"

_ijms, 2021, doi:10.3390/ijms22052739_

Round 1

Reviewer 1 Report

The perspective article by Birgitta Strandvik is titled by "Is the ENaC dysregulation in CF an effect of protein-lipid interaction in the membranes?". The interaction of CFTR and ENaC has been documented and modulation of ENaC activity was proposed to be beneficial in CF therapeutic treatment. CF patients have abnormal lipid composition which can be altered by the usage of CFTR modulators. However, there is very little evidence suggesting the changes of lipid would affect the traffic or function of ENaC. Conversely, the evidence of lipid changes due to ENaC disfunction is weak. 

Author Response

Thank you for evaluating my manuscript. I certainly agree that the evidence today is weak that lipid changes are the cause of ENaC dysfunction. That is the reason I have a question mark in the title. However, the indications are strong that the lipid abnormality is the cause of the ENaC abnormality, and those factors are the basis of the paper. If we can focus on this for further research and the presented hypothesis and already existing data can be confirmed and thereby proved, we find a cheap and for long-term treatment also safe adjuvant treatment in CF, which is highly wanted by the reason I give in the paper.

Reviewer 2 Report

This is a very interesting review/hypothesis on the possibility to explain, at least in part, the abnormal functionning of ENac in cystic fibrosis due to membrane lipid composition and plausible interaction with proteins (ENaC or CFTR).

I have several minor comments:

In the introduction

  • in ENaC, "E" stands for Epithelial (and not Endothelial)
  • check several "therapeutic" and not "therpeutic"
  • viscus circle! do you mean "vicious" or "viscous"
  • finding treatment related to lipid abnormalities "has not been encouraged"... Why? could you explain in some details
  • The three units build a central ion channel. Please use "pore" instead of "channel"

In CFTR and ENaC:

  • sphingolipids not "sfingolipids"
  •  "the the"... and next line "from from"
  • when studing, please correct this

In Linoleic acid...:

  • therpies

In conclusion:

  • please provide more info on the multicenter European clinical trial.

Author Response

Thank you for critically evaluating my manuscript, and thank you for taking into consideration the wellknown interaction between proteins in the membranes, which are built of double layer of phospholipids, which composition is known to interfere with the proteins.

Responses to your specific comments:

“E” for Epithelial – of course. I apologize for this error.

Therapeutic – of course. I again have to apologize.

I mean circulus vitiosus, e.g. vicious circle. I prefer the latin, but I thought it might not be accepted. And then I am sorry I wrote it wrongly.

This is a very intriguing question. As a matter of fact, there have been a lot of small studies with lipid supplementation, but – as in most research there is fashion also in this field. After the gene was discovered focus has rather naturally been focused on proteomics, and it has been very difficult to get founding for other kind of research in CF. Furthermore , it is difficult to make clinical double blind studies of oral supplementation. Now there is a study registered in ClinTrial.gov, as mentioned in the paper, which will run for one year strictly blindly with controls. The use of such supplementation in some centers with very good results, has not been convincingly associated with the lipid supplementation, and there has also been warnings (ref. 66 in the paper), based on cell studies that supplementation with linoleic acid should be dangerous (discussed in the paper). Personally, I also believe one reason is that oral treatment by a natural product is not “scientific” enough, and then industry is not interested because that cannot provide so much money. I think this latter is wrong, because if we can find out the mechanism, that might as well end up in a product for industrial supply.

I have changed to pore.

Sorry for the sweinglish, it should of course be sphingolipids.

Sorry for the staccato (I have a new computer and it jump in an unexpected way)

In Linoleic acid – sorry I don´t understand this point, but to make the paper uniform regarding fatty acids, I have introduced abbreviation also for this acid.

Sorry for therapies

In conclusion, more information added about the trial (which is delayed due to the pandemic).

Round 2

Reviewer 1 Report

The manuscript has been improved after modification.